# STYLEMORPH: DISENTANGLED 3D-AWARE IMAGE SYNTHESIS WITH A 3D MORPHABLE STYLEGAN

Eric-Tuan Le[1][*] Edward Bartrum[1,2][*] Iasonas Kokkinos[1]

[1] University College London  [2] Alan Turing Institute

## ABSTRACT

We introduce StyleMorph, a 3D-aware generative model that disentangles 3D shape, camera pose, object appearance, and background appearance for high quality image synthesis. We account for shape variability by morphing a canonical 3D object template, effectively learning a 3D morphable model in an entirely unsupervised manner through backprop. We chain 3D morphable modelling with deferred neural rendering by performing an implicit surface rendering of "Template Object Coordinates" (TOCS), which can be understood as an unsupervised counterpart to UV maps. This provides a detailed 2D TOCS map signal that reflects the compounded geometric effects of non-rigid shape variation, camera pose, and perspective projection. We combine 2D TOCS maps with an independent appearance code to condition a StyleGAN-based deferred neural rendering (DNR) network for foreground image (object) synthesis; we use a separate code for background synthesis and do late fusion to deliver the final result. We show competitive synthesis results on 4 datasets (FFHQ faces, AFHQ Cats, Dogs, Wild), while achieving the joint disentanglement of shape, pose, object and background texture.

## 1 INTRODUCTION

Learning the structure and statistics of the 3D world by observing 2D images is at the forefront of current vision and learning research as this can unlock applications in robotics, augmented reality and graphics while also having fundamental scientific value for advancing visual perception. In this work we aim to develop the ability to do so through a model that is highly disentangled, yielding a similar level of control to that enjoyed by 3D morphable models (3DMM) (Blanz & Vetter, 1999), without requiring anything other than an unstructured set of 2D images. 3DMMs are the workhorse of facial visual effects (VFX) in the film industry and augmented reality (AR) (Egger et al., 2021), as they provide VFX creators with fine-grained, disentangled control over expression, pose, and appearance. In this work we aspire to develop unsupervised counterparts for general object categories.

In particular we show that we can learn such models for several categories other than human faces while having no prior knowledge about the object topology or other 3D prior information or knowledge of the camera pose. We build on recent progress on 3D-aware GANs and show that we can improve the FID of the most competitive methods that use the same level of supervision (plain 2D images), while exerting more control on the image synthesis process: we disentangle shape (e.g. gender, expression, hair style), camera pose, object appearance, and background. This allows us to do fine semantic edits, that preserve all properties beyond the one we are editing. We show that this applies not only to faces but also to images of cats, dogs and and wild animals.

**Relation with previous works** *3D-aware category-level modelling* advances have shown that one can use 2D image supervision to train 3D generative models of shape and appearance variability. Starting from standard MLPs (Schwarz et al., 2020; Niemeyer & Geiger, 2021) and subsequently custom sinusoidal-based networks (Sitzmann et al., 2020; Chan et al., 2021b), 3D implicit models quickly delivered results competitive to those of voxel-based approaches (Nguyen-Phuoc et al., 2019; 2020). Hybrid models (Gu et al., 2021; Zhou et al., 2021; Or-El et al., 2021; Chan et al., 2021a;

---

Project page: https://stylemorph.github.io/stylemorph/
[*]Equal contribution.

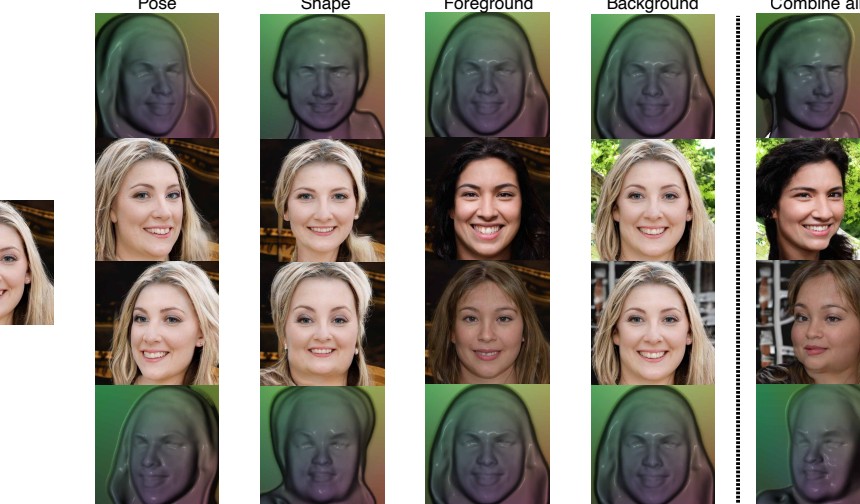

Figure 1: Our model achieves disentangled control of image synthesis: starting from a synthesized sample we change one factor at a time, and in the end show the compounded variation that we obtain by changing all. Our 3D-based conditioning signal (shown on the top and bottom rows) is exclusively geometric - it is hence the same as the left image's in the foreground and background columns, while the change is effected only by the respective appearance codes (not shown).

Xue et al., 2022) have increased the resolution and quality in which images can be synthesized without compromising speed or memory by relying on a hybrid approach that renders coarse-resolution neural features from 3D to 2D and then delegates the full-resolution image synthesis task to 2D, StyleGAN-type blocks (Karras et al., 2020). These works have shown increasingly high-quality results - but their hybrid nature makes it harder to have a clear separation of geometry and appearance or provide consistent image synthesis results when we change rigid (camera) or non-rigid (gender/expression/hair) 3D geometry.

Recent works aimed at disentangling appearance from shape have incorporated 3D deformations in the synthesis process (Park et al., 2021; Pumarola et al., 2021; Gafni et al., 2021a; Su et al., 2021; Xu et al., 2021; Weng et al., 2022), but so far have remained limited to either the single dynamic scene use case, or assume a pre-existing deformable model already exists for a category (Gafni et al., 2021a; Xu et al., 2021; Weng et al., 2022; Su et al., 2021).

In particular for faces compelling *controllable synthesis* results have been obtained by recent works that combined 3DMMs with NERFs (Athar et al., 2022; Gafni et al., 2021b) or 3D-aware GANs (Liu et al., 2022; Tewari et al., 2020). Still, constructing a 3DMM typically requires extensive 3D scanning and manual alignment, making it only meaningful for critical categories such as faces.

Most recently Tewari et al. (2022) injected 3DMMs in GAN training, showing that one can control image synthesis through 3D warps. In our work we turn 3DMMs into first-class citizens for 3D generative modelling by combining them with Deferred Neural Rendering.

*Learning 3DMMs from 2D images* has also been recently achieved for monocular 3D reconstruction based on limited information, such as binary segmentation masks (Kanazawa et al., 2018; Sahasrabudhe et al., 2019; Kokkinos & Kokkinos, 2021b), allowing us to handle a broad range of categories (Ye et al., 2021; Vasudev et al., 2022); other works have provided models that can accommodate articulation (Kulkarni et al., 2020; Kokkinos & Kokkinos, 2021a; Yang et al., 2022), and varied object topology (Duggal & Pathak, 2022), managing known shortcomings of 3DMMs. The synthesis results of these methods however rely on a parametric low-resolution surface and texture map, yielding synthetic-looking images.

**Contributions** Our work builds on advances from these three strands of research to combine 3DMMs with GANs in an unsupervised manner. We show that it is possible to inject the main idea of morphable models, i.e. deforming a fixed "canonical" template to a diverse set of "world" shapes into the design of implicit 3D networks. Existing approaches model shape variability through a random input to an occupancy network. Instead, we bridge 3D morphable models with 3D-aware

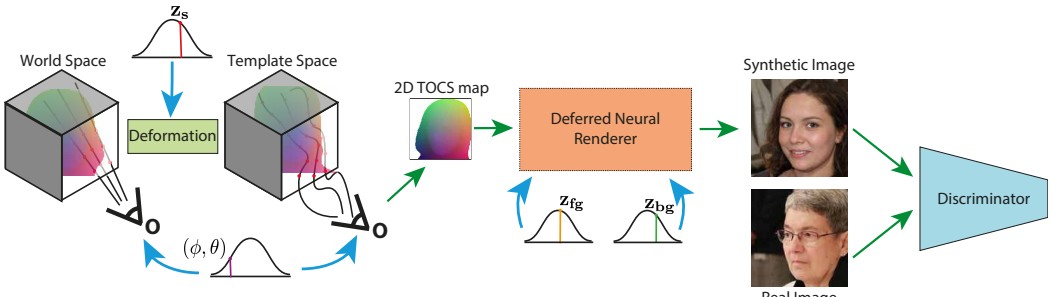

Figure 2: Overview of our approach: the left side reflects geometric modelling of template shape, non-rigid shape variation ($\mathbf{z}_s$), camera pose ($\phi, \theta$), and perspective projection, used to produce the rendered 2D TOCS that acts as a bottleneck for geometric variation. 2D TOCS maps feed into a Deferred Neural Rendering network together with latent codes for foreground and background appearance to produce high-resolution photorealistic images trained by a discriminator network.

GAN synthesis by introducing a canonical coordinate system: there, the occupancy function of the object is modelled as a constant (but learned) function. To model shape variation we sample a 3D deformation field MLP (Zheng et al., 2021) that is driven by a latent code, and use the deformation to morph the 3D canonical template thus obtaining dense correspondence between the canonical SDF space and each deformed version similarly to Palafox et al. (2021).

A main spin we introduce to existing 3D-aware models consists in rendering surface-level signals instead of RGB values or neural fields. This cleanly removes appearance information from shape modelling: the signal provided to the 2D synthesis network is purely geometric.

In particular the surface level signals we provide as conditioners are 2D maps based on the 3D "Template Object Coordinates" (TOCS) that project to a given pixel in 2D. This is inspired from the Normalized Object Coordinates (NOCS) used for 3D pose estimation in Wang et al. (2019) and allows us to bypass the challenging 2D UV parameterization of a template surface. The template occupancy function can freely evolve during training, potentially even updating its topology.

We use these TOCS maps as a proxy to 3D geometry in tandem with object appearance code to condition a StyleGAN-based deferred neural rendering (DNR) network for object (foreground) synthesis. By virtue of being defined with respect to a template, TOCS maps endow every pixel with clear semantics (e.g. indicating an eye, nose, or ear part lands on that pixel). This is reflected in our ability to produce meaningful geometric warps through the control of the underlying 3D deformation field as shown in Fig. 7.

We are also able to disentangle object (foreground) and scene (background) synthesis as in Xue et al. (2022); Chen et al. (2022) by using the object-based TOCS maps together with one appearance code to drive foreground synthesis and a separate code to account for background variability. The two synthesized images are combined with late fusion, using a TOCS-based alpha mask. This allows us to control all four sources of variability (shape, camera, object and scene appearance) separately, yielding a highly disentangled model for 3D-aware image generation, as shown in Fig. 1.

To summarise, our contributions are as follows:

- We learn a 3D morphable model of the non-rigid shape variation in an object category exclusively from 2D image supervision.

- We introduce Template Object Coordinates (TOCS), a deformable variant of Normalized Object Coordinates, and show that this provides a powerful, deformation-equivariant descriptor of 3D shape.

- We introduce TOCS maps as a powerful conditioning signal to a style-based 2D DNR, allowing a clear disentanglement of shape and appearance conditioning.

- We show unprecedented disentangled control over pose, shape, object appearance, and scene appearance for high-resolution, photorealistic image synthesis.

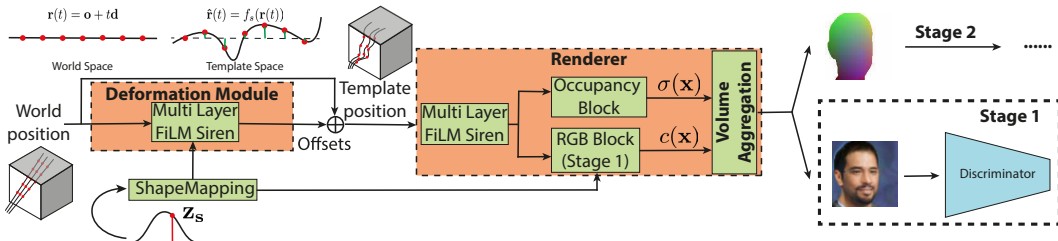

Figure 3: Architecture of the Morphable renderer: a 3D deformation field warps camera rays to template coordinates, where object occupancy and appearance are modelled more easily in a deformation-free coordinate system. In the first stage this system is trained to synthesize images (amounting to a 3D deformable variant of PiGAN) and in the second stage the learned 3D deformation model is used to provide 2D TOCS maps that condition the DNR.

## 2 METHOD

Our method takes as input an unstructured collection of RGB images of an object category. Our objective is to learn a generative model for images of the same distribution that (i) disentangles camera, shape and foreground/background appearance variation, (ii) expresses shape variability through the deformation of a learned template shape and (iii) allows us to efficiently synthesise high-resolution, realistic images.

As shown in Fig. 2. we sample separate latent codes from unit Gaussian distributions to control the various factors of variation: $z_{fg}$, $z_{bg}$, $z_s$ control foreground/background appearance, and shape respectively. We similarly sample our cameras azimuth and elevation coordinates $(\phi, \theta)$ from Gaussian distributions, to determine each pixels ray origin.

On the left side we model the effects of geometry in three steps: a deformation-free object model is learned in template object coordinates (TOCS) as a constant implicit function. This is connected to world coordinates through a 3D deformation field, represented as an implicit function driven by a latent code that accounts for non-rigid shape variability. We nonlinearly warp the camera rays from world to template coordinates and use differentiable rendering to produce "Template Object Coordinates" (TOCS), which represent the Normalized Object Coordinates (NOCS) (Wang et al., 2019) in template space. The compounded effects of camera pose, non-rigid deformation and perspective projection are reflected in this 2D TOCS map, that encapsulates all geometry information.

On the right side we have 2D StyleGAN-based Deferred Neural Rendering (DNR) network that takes as input the geometric conditioning of the 2D TOCS maps together with two separate latent codes for foreground (object) appearance and background (scene) appearance and synthesizes high-resolution photorealistic images as dictated by a discriminator network.

We call the left side a "Morphable Renderer" network and train it firstly on its own together with some auxiliary losses in order to bootstrap the whole system as detailed in Sec. 2.1. The network can be understood as a morphable variant of PiGAN (Chan et al., 2021b) obtained by injecting into the synthesis process a layer for deformable differentiable rendering. We provide more information on the second part of the network in Sec. 2.2 and conclude with specifications of the training losses and optimization procedure in Sec. 2.3.

### 2.1 MORPHABLE RENDERER BLOCK

The first part of our training pipeline, detailed in Fig. 3, is aimed at capturing all 3D geometrical aspects of image formation, including both non-rigid ("shape") and rigid ("pose") sources of variability. Following Chan et al. (2021b); Gu et al. (2021); Or-El et al. (2021) we model pose variability by positioning our camera on a unit sphere at elevation and azimuth $(\phi, \theta)$, pointing at the origin.

We generate morphs through a Gaussian shape code $z_s$ that drives a SIREN deformation model $g_s : \mathbb{R}^3 \to \mathbb{R}^3$. In particular $z_s$ predicts frequencies $\gamma_s$ and phase shifts $\beta_s$, $\gamma_s, \beta_s = \text{ShapeMapping}(z_s)$, which in turn modulate the layers of $g_s$: $g_s(p) = \text{SIREN}(p, \gamma_s, \beta_s)$. We compute the deformation field $f_s : \mathbb{R}^3 \to \mathbb{R}^3$ by adding the predicted offset to each world-space point $p$, warping them into template space: $f_s : p \longmapsto p + g_s(p)$ similarly to Park et al. (2021).

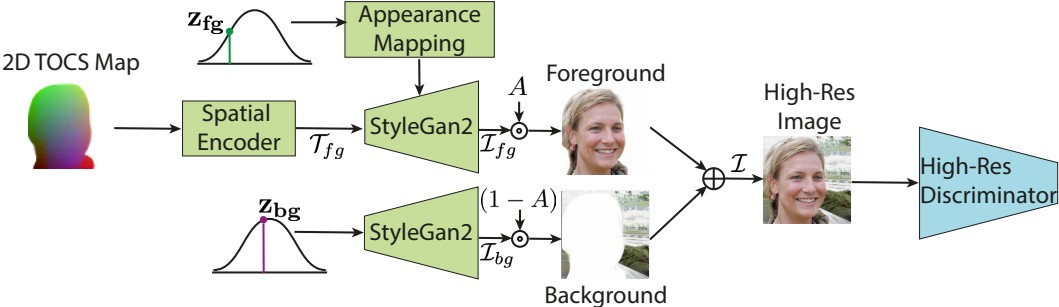

Figure 4: Architecture of the Deferred Neural Renderer: spatial features extracted from the 2D TOCS map are used to condition a StyleGan2 generator for foreground synthesis. Background, generated from a second StyleGan2 generator is composited for final synthesis.

In more detail when performing ray-tracing for each pixel, we sample world points along the corresponding ray $\mathbf{r}(t) = \mathbf{o} + \mathbf{d}t$ where $\mathbf{r}(t)$ is the world-space position of a ray sample at distance $t$ along the ray, with ray origin $\mathbf{o}$ and direction $\mathbf{d}$. Our deformable rendering layer, shown in 2D case, maps the world-space ray $\mathbf{r}$ to a deformed template-space ray $\hat{\mathbf{r}}$ by passing it through the deformation field $f_s$ defined above:

$$\hat{\mathbf{r}}(t) = f_s(\mathbf{r}(t)) = \mathbf{r}(t) + g_s(\mathbf{r}(t)). \tag{1}$$

This operation associates world points with their template pre-images and can thereby eliminate the variability caused by non-rigid object deformation. Once mapped to template space, the ray samples are passed through a second SIREN network with constant (i.e. instance-agnostic) learned frequencies and phase shifts, that represents our template implicit field.

In the *first training stage* we use this block to directly render low-resolution RGB images. We follow the SDF-based method introduced in Or-El et al. (2021) for the raysampling procedure, and convert SDF values to occupancies through a sigmoidal function ($\sigma(x)$). An appearance code drives an implicit model for RGB intensity which combined with the occupancy function yields a rendered 2D RGB image. We additionally return an estimated alpha map from our volume renderer, obtained by integrating the ray-occupancies. This allows us to obtain a 4-channel RGBA image prediction during low resolution training. We use a 4-channel low-resolution discriminator as well and for the real alpha masks we use approximate segmentations obtained using the unsupervised method of Labels4free (Abdal et al., 2021) (detailed in the appendix). This can be understood as providing weak silhouette supervision to our 3D shape model, while also priming the appearance and shape modelling to focus on the object region.

In the *second training stage* we repurpose this block to provide 2D TOCS maps to the following DNR network. In particular, when tracing a ray we now integrate the template-coordinate values $\hat{\mathbf{r}}(t)$ along the ray to obtain the TOCS value:

$$\mathrm{TOCS}(\mathbf{r}) = \int_{t_n}^{t_f} w(\hat{\mathbf{r}}(t))\hat{\mathbf{r}}(t)\mathrm{d}t \tag{2}$$

where $\mathrm{TOCS}(\mathbf{r})$ is the 2D TOCS map value corresponding to ray $\mathbf{r}$, $w(\hat{\mathbf{r}}(t))$ is the SDF-based weight computed at the template point $\hat{\mathbf{r}}(t)$ as in Or-El et al. (2021) and $t_n$ and $t_f$ are the near and far ray limits, which are fixed across training; we approximate the integral using discrete sampling.

We note that during the DNR training, rather than using an appearance or radiance function $C(\hat{\mathbf{r}}(t))$ as an argument to the weighted integral in Eq. 2, we directly use the ray's template coordinate position $\hat{\mathbf{r}}(t)$, amounting to the TOCS representation; if instead of $\hat{r}(t)$ we were using $r(t)$ this would be providing us with the standard 2D NOCS map. NOCS maps prove to be strong conditioning signals for DNR, which by itself is a new and interesting result, but as our experiments show, there is a substantial improvement in FID scores by using TOCS instead of NOCS through the deformation field while obtaining dense shape correspondence for free.

We limit the resolution of the rendered TOCS map to $64^2$ (regardless of final image resolution) as this keeps the memory and computation footprint of this stage low, while sufficing for detailed pose and shape conditioning. We also do not use the RGB rendering blocks during the second stage of training or the final inference process.

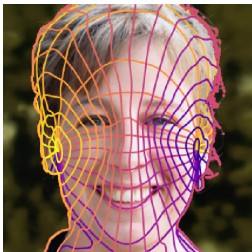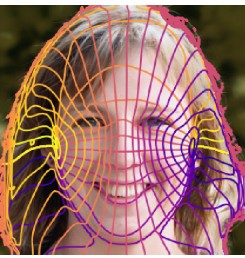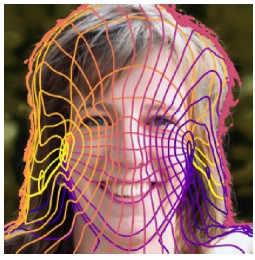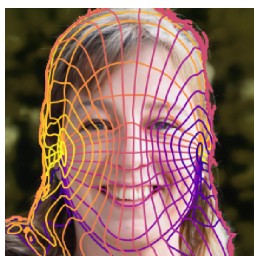

Figure 5: From a fixed viewpoint, we render TOCS maps with 4 different shape codes, and pass them to the DNR with the same background and foreground appearance code. The Morphable Renderer generates diverse TOCS maps, corresponding to different hair styles and face shapes, which are with strong alignment with the synthesised RGB. We visualise the template coordinates using equicontours, to show the surface correspondences between shapes.

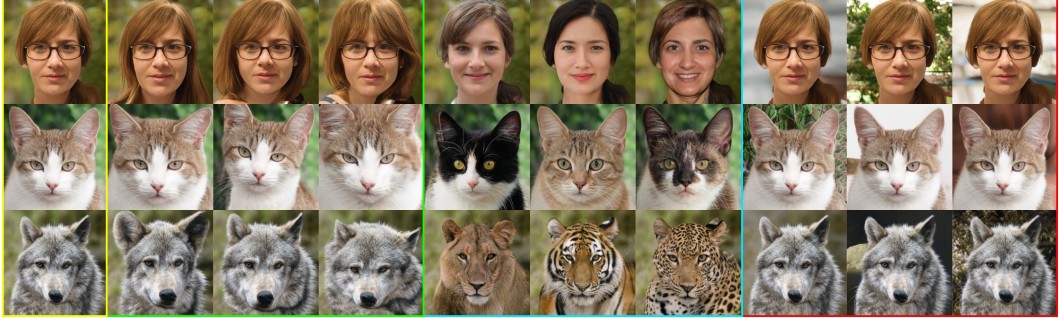

Figure 6: Starting from a source sample (yellow border) we demonstrate disentangled appearance control over shape (green border), foreground (blue border) and background (red border).

## 2.2 TOCS-CONDITIONED DNR

Our Deferred Neural Rendering pipeline, shown in Fig. 4, follows recent works on UV-driven Style-Gan networks for human synthesis (Sarkar et al., 2021; AlBahar et al., 2021) or segmentation-driven synthesis (Park et al., 2019). First we process the rendered TOCS map with a small residual network (Spatial Encoder in Fig. 4) that preserves the original spatial resolution, but transforms the 3-D TOCS values to a richer, 64-D representation. The resulting foreground feature tensor $\mathcal{T}_{fg}$ replaces the early blocks of a StyleGAN generator, providing a pose conditioning signal that captures the joint effects of shape and camera variables. Our TOCS map contains informative geometry-based values only on the surface of the deformed shape (which we interpret as the foreground region). Since we wish to generate detailed backgrounds, we use a separate 2D Background Generator to create a background RGB image $\mathcal{I}_{bg}$. Our Background Generator consists of a small StyleGAN generator, which outputs a full-resolution background image. $\mathbf{z_{bg}}$ is passed to the mapping network of the Background Generator, so that it controls the content of the background.

Our foreground generator similarly consists of StyleGAN blocks, which are used to upsample the low resolution feature tensor $\mathcal{T}_{fg}$ to a full resolution RGBA tensor. This contains the foreground RGB image $\mathcal{I}_{fg}$, and an upsampled alpha map $A$. Similarly to the background case, $\mathbf{z_{fg}}$ is used to modulate the activations of the DNR StyleGan blocks through Adaptive Instance Normalization, via a mapping network.

We composite the foreground RGB $\mathcal{I}_{fg}$ with the background RGB $\mathcal{I}_{bg}$ using alpha blending with the upsampled alpha mask $A$ of the foreground generator: $\mathcal{I} = A \odot \mathcal{I}_{fg} + (1 - A) \odot \mathcal{I}_{bg}$

## 2.3 NETWORK TRAINING

We now provide more information on the process followed for training our system - we provide more details in the appendix and will share code for reproducibility.

**Training strategy:** We adopt a 2-stage low-to-high resolution training strategy following Or-El et al. (2021) to limit memory footprint and keep a decent batch size for optimal overall performance and speed. For stage 1, our deformable volume renderer (incorporating template and deformation-

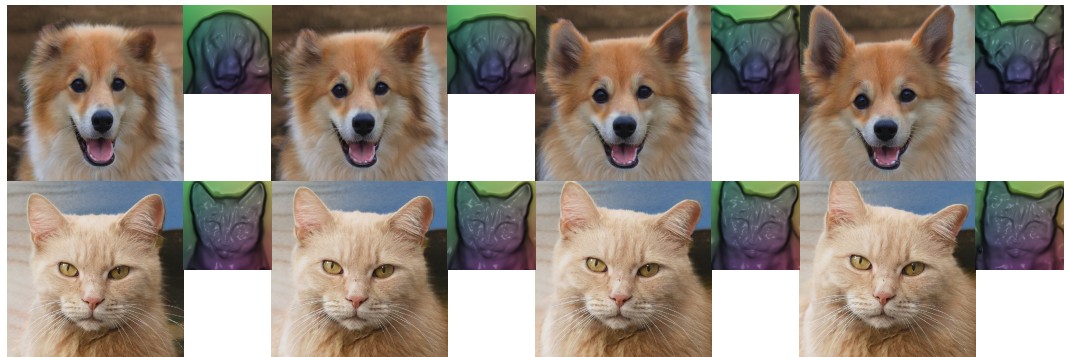

Figure 7: Deformation results on AFHQ Dogs and Cats. We linearly interpolate shape instances through deformation-offset space. The deformation field captures complex non-linear motion of the dog's ears, and relative motion of the cat's head to the torso.

offset networks) is trained as a low-resolution ($64^2$) image generator, in order to learn a realistic shape model. To generate low-resolution training images without the use of the DNR, we include an RGB prediction block in the implicit field following Schwarz et al. (2020); Chan et al. (2021b); Or-El et al. (2021), which is not necessary for the full-resolution synthesis.

In stage 2, we freeze the volume renderer weights, and train the DNR as a full-resolution 2D generator conditioned on the projected TOCS maps. We prune the RGB synthesis block from the volume renderer as it is no longer needed. We use the volume renderer only to render TOCS maps for given poses and shape codes, and train our DNR with appearance conditioning purely in the 2D domain. The DNR can be trained at arbitrary resolutions conditioned on the $64^2$ TOCS maps.

**Loss functions:** In both cases we apply the non-saturated GAN training objective (Mescheder et al., 2018) with R1 regularisation and path regularisation to force our generator to learn to synthesise realistic images, which we denote as $\mathcal{L}_{gen}$.

During the first stage of training, we additionally regularise the volume renderer using shape losses to stabilize optimisation and avoid degenerate local minima, as detailed in the appendix.

For the 2nd phase of training, we fix the volume renderer and drop shape losses. We additionally use an L1 loss $\mathcal{L}_{alpha}$ to ensure consistency between the low-resolution and upsampled alpha maps. The generator training objective is therefore $\mathcal{L}_{gen} + \mathcal{L}_{alpha}$. We report the reader to Appendix 11.1.3 for explicit loss equations.

## 3 EXPERIMENTS

### 3.1 DATASETS

We evaluate our pipeline together with 11 state-of-the-art baselines on the FFHQ (Karras et al., 2019) and AFHQ (Choi et al., 2020) datasets. FFHQ is composed of $70k$ centred images of human faces, with a variety of challenging backgrounds and poses. AFHQ is composed of centred images of animal faces, split into 3 categories: 5653 Cats, 5239 Dogs and 5238 Wild. We report FID results on FFHQ and all AFHQ datasets to evaluate the quality of our model's 2D synthesis.

### 3.2 QUANTITATIVE EVALUATION

**Baselines:** We compare our synthesis results to several state-of-the-art 3D-GAN models by FID score in Table. 1. We group the recent photo-realistic stylegan-based methods together at the bottom. Our FID scores are competitive with the state-of-the-art models, despite our additional disentanglement and template-based shape model constraints, which can in principle compromise the generator's flexibility. The only directly comparable method is GiraffeHD (Xue et al., 2022) which however still lacks an underlying template coordinate system (hence making it hard e.g. to transfer masks defined on a template, as shown in our Appendix). The only prior 3D-GAN work which represents shape via a deformable template is the independently developed Disentangled3D work (Tewari et al., 2022). However, this significantly underperforms ours on the FFHQ dataset in Ta-

| | FFHQ $256^2$ FID | AFHQ $256^2$ FID | | | | Disentanglement | | Template | Unposed |
|---|---|---|---|---|---|---|---|---|---|
| | - | Cat | Wild | Dogs | Joint | Shape | Scene | - | - |
| HoloGAN | 75.00 | - | - | - | 78.00 | ✓ | ✗ | ✗ | ✓ |
| GRAF | 71.00 | - | - | - | 121.00 | ✓ | ✗ | ✗ | ✓ |
| GIRAFFE | 31.20 | - | - | - | 31.00 | ✓ | ✓ | ✗ | ✓ |
| pi-GAN | 34.56 | 38.92 | - | - | - | ✗ | ✗ | ✗ | ✓ |
| GRAM | 29.80 | - | - | - | - | ✗ | ✗ | ✗ | ✗ |
| Disentangled3D | 28.18 | - | - | - | - | ✓ | ✗ | ✓ | ✓ |
| StyleNERF | 8.00 | - | - | - | 14.00 | SG-Based | ✗ | ✗ | ✓ |
| CIPS3D | 6.97 | - | - | - | - | ✓ | ✗ | ✗ | ✓ |
| StyleSDF | 11.50 | - | - | - | 12.80 | ✗ | ✗ | ✗ | ✓ |
| EG3D* | 4.80 | 3.88 | - | - | - | SG-Based | ✗ | ✗ | ✗ |
| GiraffeHD | 11.93 | 12.36 | - | - | - | ✓ | ✓ | ✗ | ✓ |
| Ours | 7.91 | 4.29 | 3.49 | 13.95 | - | ✓ | ✓ | ✓ | ✓ |

Table 1: Comparisons with the state-of-the-art on 3D-aware GANs; top block: direct 3D methods; bottom block: 3D-2D hybrids. We note that (a) EG3D uses a pre-existing 3D pose estimation network, hence is using weak 3D supervision (unlike other methods) (b) only Disentangled3D and GiraffeHD are comparable to us in terms of disentanglement. We provide more details in the text.

ble. 1 due to the absence of StyleGAN synthesis blocks. Furthermore, this work does not address foreground/background synthesis.

**Ablation:** To justify our architectural choices, we perform an ablation study in Table. 2. We use multiple more metrics beyond the ones used in Table. 1, as these allow us to better understand the tradeoffs for our different design choices. A full discussion of consistency scores can be found in the supplemental material. For fair apples-to-apples comparison, we compare all models after equal training time (96 hours) on equivalent hardware.

Our proposed method is in row 1 - Late fusion, and differentiable TOCS rendering. The first check we perform in row 2 is what happens if rather than TOCS we render NOCS, while using the exact same system - i.e. the 3D world coordinate of a point, rather than its 3D template coordinate. We observe decreased performance across almost all metrics. This is a sign that TOCS helps both generate sharper samples, and ensure disentanglement in synthesis - while NOCS provides a more fuzzy signal, since it lacks that clarity of template-level conditioning.

Row 3 repeats this experiment but now removing the deformable component and replacing it with a standard implicit model for occupancy driven by a shape code. This reduces a bit the FID score and increases appearance variability wrt Row 2, but drastically worsens consistency scores. This indicates the added value of a deformable module to model shape variation as opposed to doing direct occupancy estimation through an MLP - the difference is more pronounced wrt Row 1.

Row 4 finally shows how results change when we have early fusion. We have a slightly smaller FID (potentially due to the looser earlier fusion which gives the generator more flexibility) but clearly worse alpha consistency scores - indicating that the synthesis results can override the conditioning TOCS map and blend foreground with background.

A last ablation, shown in Row 5a examines if we can improve the view consistency of our model, by directly optimising the view-consistency metric during training by formulating it as a differentiable loss. We observe a trade-off between view-consistency and image quality, as optimising view-consistency causes a corresponding increase in FID score. Nevertheless, we find that we can obtain a similar FID score to StyleSDF with a significantly lower view-consistency.

## 3.3 QUALITATIVE EVALUATION

Our full pipeline model generates high-resolution, photorealistic images. We observe in Fig. 5 that the Morphable Renderer can generate diverse shapes capturing the complex shape variability present in the FFHQ dataset. Our approach allows us to factor shape variability into canonical template instance and shape-specific deformation, directing the full capacity of the network towards capturing intra-category shape variation. Furthermore, we note in Fig. 5 that DNR synthesis re-

| Id # | Ablations | | | | Perf. | Consistency | | | | Variation |
|---|---|---|---|---|---|---|---|---|---|---|
| | FG/BG Compositing | Morph | NOCS / TOCS | Reproj. Loss | FID↓ | View↓ | Alpha↑ | Shape↓ | Appearance↓ | Appearance↑ |
| 1 | Late | ✓ | TOCS | | 8.309 | 15.843 | 88.65% | 0.798 | 0.015 | 0.093 |
| 2 | Late | ✓ | NOCS | | 8.90 | 17.493 | 89.4% | 0.810 | 0.018 | 0.078 |
| 3 | Late | | NOCS | | 8.531 | 19.026 | 89.85% | 0.807 | 0.016 | 0.089 |
| 4 | Early | ✓ | TOCS | | 8.186 | 13.638 | 85.87% | 0.870 | 0.018 | 0.079 |
| 5a | Late | ✓ | TOCS | ✓ | 12.31 | 8.12 | 89.90% | 0.815 | 0.015 | 0.085 |
| 5b | StyleSDF | | | | 11.5 | 13.60 | - | - | - | - |

Table 2: Quantitative comparisons of ablations; please see text for details

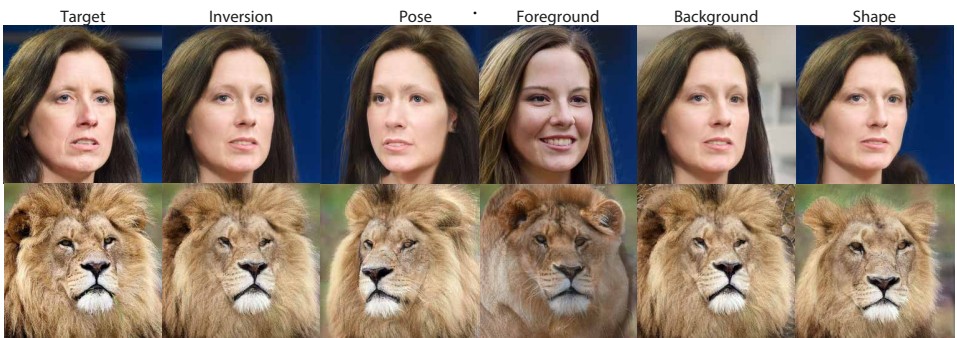

Figure 8: StyleMorph-inversion: by inverting our pipeline, we can reconstruct a real input image with its underlying 3D structure and exert full control over multiple sources of disentanglement.

mains closely aligned to the TOCS map, generating photo-realistic fine details in the hair and eyes without deviating from the coarser geometric structure. The strong alignment exhibited by the DNR between TOCS values and RGB, combined with the dense semantic correspondences between projected TOCS maps, enables the deformation-equivariant AFHQ synthesis results shown in Fig. 7. We synthesise images conditioned by source shape instances on the left, then linearly scale the deformation field values up to the target shape instances shown on the right. The resulting non-linear motion in the TOCS maps is tracked equivariantly in the RGB synthesis, as seen in the raising of the dogs ear and twisting of the cat torso.

As detailed in Sec. 2, our approach uses separate latent codes for shape, background and foreground appearance. By varying codes in one latent space and holding the others fixed, our model exhibits disentangled synthesis across each factor of variation, and all datasets. We observe in Fig. 6 that the deformation network can change the woman's hair style, and twist the cats head relative to its torso. The background generator can synthesise a variety of structures without impacting the foreground, whilst varying the foreground code results in diverse appearances, whilst maintaining foreground/background consistency.

We show in Figure 8 how a real image can be inverted using Pivotal Tuning (Roich et al., 2021) to recover the 3d model from a single image. Thanks to our disentangled synthesis, the inital input image can be edited via multiple sources of control: pose, foreground, background and shape.

# 4 CONCLUSION

In this work we have shown that 3D deformation-based control over image synthesis can be achieved without any compromise to state-of-the-art 2D synthesis quality. We provide more results and videos in the Appendix and will make our code publicly available.

**Acknowledgements** All final models described in this research were trained using the Baskerville Tier 2 HPC service (https://www.baskerville.ac.uk/). This was funded by EPSRC Grant EP/T022221/1 and is operated by Advanced Research Computing at the University of Birmingham. This work was also supported in part by JADE: Joint Academic Data science Endeavour - 2 under the EPSRC Grant EP/T022205/1, & The Alan Turing Institute under EPSRC grant EP/N510129/1. We are grateful to Gabriel Browstow and Niloy Mitra for providing useful discussions and insights.

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
