# OpenReview forum: "StyleMorph: Disentangled 3D-Aware Image Synthesis with a 3D Morphable StyleGAN"
_ICLR.cc/2023/Conference — ICLR 2023 poster_

### Official Review · Reviewer_cRkf · 2022-10-22

**Confidence:** 4
**Correctness:** 2
**Technical Novelty And Significance:** 2
**Empirical Novelty And Significance:** 2
**Recommendation:** 3

**Clarity, Quality, Novelty And Reproducibility:**

 - Clarity: 5 out of 10. I think the clarity of the method part is okay, which tells what it is doing. But the introduction is exaggerated.
 - Quality: 6 out of 10. As I have mentioned in the Strength And Weaknesses, the image quality is good, while the experiments are not sufficient. And the writing makes me upset.
 - Novelty: 1 out of 10. I see some similarities with previous works. Please let me know if I misunderstood.
 - Reproducibility: 6 out of 10. No code was submitted, but they provide details including hyper-parameters and loss functions. So I think it could be reproduced given enough time.

**Strength And Weaknesses:**

Strength:
1. The image quality in the paper looks good.
2. The supplementary materials are detailed.

Weaknesses:
1. Morphable Renderer lacks novelty. The main idea of the Morphable Renderer (disentangling the geometry and appearance + warping the learned template space to modify the geometry) is too similar to the Disentangled3d[1]. Actually, in my opinion, they are almost the same.
2. Deferred Neural Render lacks novelty. The proposed DNR takes the TOCS (a sort of geometry representation) as input and yields a corresponding high-resolution image, which is similar to StyleNeRF[2], SofGAN[3], etc. and has little novelty from my perspective.
3. The experiments are not sufficient.
    - Efficiency of TOCS. Although the TOCS outperforms NOCS, there lacks a comparison with some naive methods, e.g. directly feeding the Deferred Neural Render with the low-resolution image/depth outputs of Morphable Renderer, or using an on-the-shelf super-resolution model, e.g. GFPGAN[4], to enhance the image quality.
    - No expression control experiments.
4. The writing is exaggerated and not clear:
    - "we disentangle shape (e.g. gender, expression, hair style), camera pose, object appearance, and background". No gender or expression disentanglement was found. And the background disentanglement here is trivial, where they use a pretrained 2D generator to synthesise the plain background. It is strange to see the background keeping static while the foreground is rotated, as shown in the YouTube video.
    - "yielding a similar level of control to that enjoyed by 3D morphable models (3DMM)". I think this paper cannot control facial expressions as in the 3DMM. And the warping field is hard to edit.
    - "show that we can improve the FID of the most competitive methods". I did not find experiment results to show this general improvement.
    - "changing 3D expression, pose etc". Did not find experiment results.
    - "but their hybrid nature makes it harder to have a clear separation of geometry and appearance". why? The hybrid architecture does not conflict with the disentangled design.
5. Appearance-geometry misalignment. In Fig.1 in the main paper, the geometry of column "foreground" and column "background" is the same, which suggests the corresponding images have the same shape. However, when looking at the images, you will notice a change of expression when adjusting the appearance, indicating the appearance and geometry are not aligned well semantically.

[1] Ayush Tewari, Xingang Pan, Ohad Fried, Maneesh Agrawala, Christian Theobalt, et al. Disentangled3d: Learning a 3d generative model with disentangled geometry and appearance from monocular images. arXiv preprint arXiv:2203.15926, 2022. \
[2] Jiatao Gu, Lingjie Liu, Peng Wang, and Christian Theobalt. Stylenerf: A style-based 3d-aware generator for high-resolution image synthesis. arXiv preprint arXiv:2110.08985, 2021. \
[3] Anpei Chen, Ruiyang Liu, Ling Xie, Zhang Chen, Hao Su, and Jingyi Yu. Sofgan: A portrait image
generator with dynamic styling. ACM Transactions on Graphics (TOG), 41(1):1–26, 2022. \
[4] Wang, Xintao, et al. "Towards real-world blind face restoration with generative facial prior." Proceedings of the IEEE/CVF Conference on Computer Vision and Pattern Recognition. 2021.

**Summary Of The Paper:**

This paper introduces StyleMorph, a high-resolution 3D-aware generative model with morphable geometry and disentangled appearance.
The network is composed of two components, where a Morphable Renderer is trained to yield the TOCS map, and a Deferred Neural Renderer learns to translate the TOCS map to a high-resolution image.

**Summary Of The Review:**

In general, I think this paper has good qualitative results, but lacks novelty, and the writing is not clear and objective enough.

---

> ### Author Response · Authors · 2022-11-19
> **Response to reviewer cRkf part 1/3**
>
> **R: “Morphable Renderer lacks novelty. The main idea … is too similar to the Disentangled3d[1]. Actually, in my opinion, they are almost the same.”**
>
> Answer: These points lack nuance. Assuming the two methods are "almost the same", or the same, as a Novelty score of 1/10 suggests, this does not explain how our FID is 7.91 while that of D3D is 28.18 (Paper Table 1).
>
> This is not a difference of one or two FID points (which we agree could be due to tuning) - but rather makes the difference between providing proof-of-concept deformations and showing they can serve as a first-class citizen for image synthesis.
>
> Reviewer cRkf should indicate alternative published works that can make this happen while retaining the same level of disentanglement - otherwise we argue the critique is unsubstantiated.
>
> Beyond D3D there are many other works that try to bridge morphable models with image generation - we had already cited originally and now added more to the prior work section. We regret not discussing D3D in the prior work section of the original submission - this was an omission due to time pressure - but we had already included and compared to it in our experimental results.
>
> In particular Disentangled3D can be seen as a variant of our lower branch in Figure 3. This is only a part of our system - as our paper title states, we rely on "Style" to generate high-res images, i.e. our DNR block. Combining the two pieces together is non-trivial and took experimentation and thought -- over one and a half years, since we were working independently of D3D.
>
> As mentioned originally in the intro: “A main spin we introduce to existing 3D-aware models consists in rendering surface-level signals instead of RGB values or neural fields. This cleanly removes appearance information from shape modelling: the signal provided to the 2D synthesis network is purely geometric.”. This is entirely new, and constitutes the link between morphable models and StyleGan (hence StyleMorph as our paper’s title).
>
> **R: “Deferred Neural Render lacks novelty. The proposed DNR takes the TOCS (a sort of geometry representation) as input and yields a corresponding high-resolution image, which is similar to StyleNeRF[2], SofGAN[3], etc”**
>
> Answer: Firstly, “novelty” is highly subjective. TOCS is entirely new, and it took us time to figure it out (it seems simple, but after we described it). In particular  we designed TOCS from scratch as a differentiable, more general layer to replace the stringent requirements of UV conditioning used eg. for DensePose-conditioned synthesis. This allows us to work with an evolving “template” and does not require a UV chart. Please provide us with similar works and we will be glad to compare or even withdraw if this has been done before.
>
> These are indeed many other geometry layers - but they are not comparable. Note that SofGAN [3] requires multiview semantic segmentation maps and 3D portrait scans in order to train (Figure 3,[3]).
>
> In contrast, our TOCS-based approach (Paper Figure 2) allows unsupervised controllable synthesis, and is therefore applicable to general image categories, not just human faces. Furthermore, it provides dense correspondence from shape instances to the canonical template space, rather than just semantic region labels as in [3]. As we show in the Disentanglement columns of Paper Table 1, StyleNeRF does not directly address disentanglement.
>
> **R: “Experiments are not sufficient. Although TOCS outperforms NOCS, there lacks a comparison with some naive methods, e.g. feeding the DNR with the low-resolution image/depth outputs of Morphable Renderer, or using an on-the-shelf super-resolution model, e.g. GFPGAN[4]”**
>
> Answer: We agree more ablations could help - but to us it was clear that the proposed baselines would be weak (or “naive” in the Reviewer’s words), so we only compared to NOCS as a strong baseline. We now provide these ablations in Supplemental Table 2 (updated supplemental).
>
> We find that both the depth and low-res RGB are less descriptive DNR conditioners than the TOCS maps, resulting in substantially worse FID scores of 20.19, 25.08 respectively. Furthermore, passing our low-resolution rgb samples to an FFHQ-pretrained GFPGAN [4] (version 1.3) results in FID 55.64. This score is in line with the FID scores reported in [4] of 42.62 on CelebA blind-face image restoration. These ablation studies demonstrate that our TOCS-based DNR is a critical component for the high-quality image-synthesis capability of our model.
>
> | Full model | Depth map | Low-res RGB | GFPGAN[4]|   |
> |------------|-----------|-------------|-----------------------------------|---|
> | 8.31       | 20.19     | 25.08       | 55.64                             |   |

---

> > ### Author Response · Authors · 2022-11-19
> > **Response to reviewer cRkf part 2/3**
> >
> > **R: “Writing is exaggerated and not clear. ‘we disentangle shape (e.g. gender, expression, hair style), camera pose, object appearance, and background’. No gender or expression disentanglement was found.”**
> >
> > Answer: We did not state that we disentangle images into these distinct sources of variation or even suggested we disentangle them individually.  This is a wrong conclusion based on misinterpreting our statement.
> >
> > In particular we state that we disentangle into “foreground, background, pose, shape (e.g.expression, gender),”  and not that we disentangle into “foreground, background, pose, expression, gender,” as cRkf’s conclusion would suggest.
> >
> > Expression and gender are listed as indicative specifications of what we mean by “shape”; we could say "non-rigid 3D surface variation" - but we find ``shape'' easier to communicate as a term, and "expression, gender" useful as concrete examples to ML readers who may not have a  Vision or Graphics background. We believe our phrasing is unambiguous and consistent with all the messaging and results in the rest of the paper, hence do not change it in our abstract and intro. Still, we removed a statement about “pose and expression control being possible on cats and dogs”, since it indeed could suggest that we have already done it.
> >
> > In particular 3DMMs require expression/gender labels to learn and manipulate 3D shape changes for those attributes (beyond 3D scans); learning those in an unsupervised way is beyond scope of this work.
> >
> > Having said this, we agree that some indicative results for shape-driven expression and gender control can motivate combining morphable models with DNR - even though we do not want to claim our method does this. Spurred by the feedback of cRkf we  provide  qualitative results in Supplemental Figure 5 (updated supplemental), using our FFHQ model to demonstrate expression control via deformation interpolation. Additionally, we train our model on the AffectNet-Sample kaggle dataset, a dataset that is richer in facial expression variation, and show similar qualitative results for that model in Supplemental Figure 6. We also show shape codes corresponding to male/female samples in Supplemental Figure 7, for our FFHQ model and caricature generation in Supplemental Figure 9.
> >
> > **R: “The background disentanglement here is trivial, where they use a pretrained 2D generator to synthesise the plain background….”**
> >
> > Answer: This is wrong: we do not use a pretrained 2D generator. As we had explained in the originally submitted paper (Section 2.2 and  Figure 4) the background RGB generator is learned alongside the foreground generator during stage 2 training.
> > Maybe cRKf confounds this with us using Labels4free for the segmentation masks?
> >
> > **R: “It is strange to see the background keeping static while the foreground is rotated”**
> >
> > Answer: We learn a generative model of 3D controllable foregrounds with disentangled 2D backgrounds (Paper Section 2.2). We did not try modeling the background as a 3D morphable field - which we believe does not make sense - while generic 3D modeling of the background would be a gimmick in our case (irrelevant to the paper’s contribution, which is category-specific). The main thing we achieve through background modeling is not requiring masked images as D3D does and training high-end discriminators that can look into the combination of faces with their backgrounds.
> >
> > **R: “‘I think this paper cannot control facial expressions as in the 3DMM. And the warping field is hard to edit.”**
> >
> > Answer: It is not clear what “hard to edit means”  - we have shown several examples of editing the warping field in Figure 8 of the main paper and Figure 2 of the supplemental - even for the case of a Lion’s head.
> >
> > Regarding facial expressions: we are saying it is “similar” level of control and not “the same”- it is absurd to expect that an unsupervised model will be as powerful as a strongly supervised one, also using additional labels for gender, expression etc - but we still need to somehow explain that this can be used in graphics and AR/VR in the same way that a morphable model has been used for 20 years. Our editing results in Figure 8 (main paper) / Figure 2 (supplemental) show this is the case, and we are not aware of similar results having been obtained in the literature before us with an unsupervised method.

---

> > > ### Author Response · Authors · 2022-11-19
> > > **Response to reviewer cRkf part 3/3**
> > >
> > > **R: "’show that we can improve the FID of the most competitive methods’. I did not find experiment results to show this general improvement.”**
> > >
> > > Answer: As we show in Paper Table 1, our model outperforms the FID scores of GIRAFFEHD (the current SOTA disentangled 3D-aware GAN) and Disentangled3D (the current SOTA deformation-based 3D-aware GAN) by considerable margins. These are the most competitive methods when it comes to disentanglement (even though as shown by our table, we distentangle more factors).
> > >
> > > **R: “The hybrid architecture does not conflict with the disentangled design.”**
> > >
> > > Answer: The usage of neural features predicted by a shared geometry/appearance backbone implicit network allows mixing of appearance and shape information in the trained network weights, as we show in Supplemental Section 7 (updated supplemental) and Supplemental Figure 4 (updated supplemental). In contrast, our approach uses a purely geometric conditioning signal (TOCS map, Paper Figure 2) as input to the stylegan blocks. As explained in Paper Section 1, “this cleanly removes appearance information from shape modelling: the signal provided to the 2D synthesis network is purely geometric.”
> > >
> > > **R: “Fig.1: Appearance-geometry misalignment …change of expression when adjusting the appearance indicates the appearance and geometry are not aligned well semantically.”**
> > >
> > > Answer: It is true that geometric consistency is not perfect -  as also reported in Table 1 of the supplemental material, this is indeed the single metric where our method underperforms competing methods. We hope the remaining four metrics, alongside with the FID scores also count in favor of our approach (5 out of 6 metrics). As also shown in Figure 1, the change of expression is minor.
> > >
> > > We note that when we were experimenting with models that gave higher FIDs our geometric consistency scores were better - this suggests that we can easily be consistent by avoiding any interesting details (e.g. by having a uniform, featureless texture) -this appears to be the case also with Distentangled3D, where a very high FID comes with strong geometric consistency.

---

### Official Review · Reviewer_tGFV · 2022-10-24

**Confidence:** 3
**Correctness:** 4
**Technical Novelty And Significance:** 3
**Empirical Novelty And Significance:** 3
**Recommendation:** 8

**Clarity, Quality, Novelty And Reproducibility:**

The paper is written in a clear way and seems to provide sufficient details for reproducibility. The authors promised to release their code. The main idea smartly combines multiple existing components and introduces a notion of the TOCS used to condition the DNR.

**Strength And Weaknesses:**

# Strengths
While the approach builds on many existing components, it combines them in a creative way to achieve a disentangled representation of the camera/shape/appearance while being able to synthesize compelling images. The method is unsupervised.
The paper is very well structured and written in a comprehensive way where it is easy to follow the core ideas.
The experiments are compelling and the authors justify their design choices through a thorough ablation study.

# Weaknesses
## Related Work
While the Related Work section is comprehensive, I would suggest that the authors still consider reviewing and citing the following two publications, which use very similar concepts as the authors of this paper. Specifically, Nerfies [1] makes use of the concept of deforming the camera rays to support a deforming scene and NPMs [2] employ a pose network which also uses a deformation field to express the correspondence between a canonical SDF space and its deformed (posed) version.
Probably should cite Nerfies and NPMs as both use the similar idea of warping the camera rays and learning the deformation field of a template.

## Methodology
- Fig. 4: It would help the reader if the variables from the text of section 2.2 (e.g. T_{FG}, I_{BG} etc.) were indicated as the inputs/outputs of the individual blocks in the image.
- Section 2.1: It would help the reader to provide the intuition behind how w() of Eq. 2 is computed and what does it represent. Similarly, the concept of TOCS could be very briefly introduced directly in the Methodology on top of linking to the paper [Wang et al. 2019] which introduced the related NOCS.
- Section 2.3: Similarly to the comment above, it would help the reader if the authors explicitly noted the mathematical formulas for the used loss function.
- Fig. 3: The figure indicates that the Shape and Appearance Mapping modules share weights but this is never described in the text. Could the authors elaborate?

## Typos:
- pg. 3: "encapsulates all geometr information." -> "encapsulates all geometry information."
- pg. 4: "In a first/second stage" -> "In the first/second stage"

## Questions for the authors
Since the 2D TOCS map is immediately passed through a spatial encoder (which produces a 64D feature vectors), one could imagine that the network is not really incentivized to produce a meaningful 3D shape (encoded by the 2D TOCS). The Figure 7 indicates that the TOCS are reasonable shapes given the corresponding RGB images, but it is also clear that there is quite some high-frequency noise on the surface geometry. Could the authors comment on why this is happening? Alternatively, have the authors experimented with regularizing the shape which the TOCS represent (to me closer to a more plausible 3D shape) and what were the observations?

- [1] K. Park. et al. Nerfies: Deformable Neural Radiance Fields. ICCV'21
- [2] P. Palafox. et al. NPMs: Neural Parametric Models for 3D Deformable Shapes. ICCV'21


**Summary Of The Paper:**

The authors present a method to synthesize photo-realistic images of objects from a given category while the approach allows for disentangled representation in terms of the camera pose, crude object shape, and foreground/background appearance. The approach introduces the concept of template object coordinates (TOCS) which both sidestep the need to perform 3D shape UV unwrapping and also allow for representing deformed instances of the shape in the common canonical form. The authors show competitive results while allowing for more semantic control over the synthesized images.

**Summary Of The Review:**

Thanks to the creativity of the method, compelling results and good quality of the presentation I am leaning towards acceptance. There are a couple of minor remarks which I would like the authors to comment on (see the weaknesses section).

---

> ### Author Response · Authors · 2022-11-19
> **Response to reviewer tGFV**
>
> **R: “Probably should cite Nerfies and NPMs”**
>
> Answer: We thank the reviewer, and we have updated the manuscript to include the suggested related works. We have also cited D-Nerf and also added further citations to other works bridging morphable models and generative model training.
>
> **R: “Fig. 4: It would help the reader if the variables were indicated”**
>
> Answer: We have updated the Paper Figure 4 accordingly for better clarity.
>
> **R: “Section 2.1: Provide the intuition behind w() of Eq. 2 …”**
>
> Answer: Please see Supplemental Section 13.1.2 in the updated supplemental.
>
> **R: “…concept of TOCS could be very briefly introduced directly in the Methodology on top of linking to the paper [Wang et al. 2019] which introduced NOCS.”**
>
> Answer: We have updated the Methodology (Paper Section 2) of the manuscript to briefly introduce the TOCS as the Object Coordinates in the template space.
>
> **R: “Section 2.3: … explicitly note the mathematical formulas for the used loss function.”**
>
> Answer: We have updated this section, pointing to the loss function formulas in Supplemental 13.1.4, updated supplemental.
>
> **R: “Fig. 3 indicates that the Shape and Appearance Mapping modules share weights … could the authors elaborate?”**
>
> Answer: For the sake of clarity, we have simplified Paper Figure 3. During phase 1 training, a single mapping network is used for both shape and appearance.
>
> **R: “typos on pg. 3: geometr -> geometry. pg. 4: In a first/second stage -> In the first/second stage"**
>
> Answer: We thank the reviewer for reporting the typos. Both are fixed in the current manuscript.
>
> **R: “Since TOCS is immediately passed through a spatial encoder … network is not really incentivized to produce a meaningful 3D shape (encoded by the 2D TOCS). …”**
>
> Answer: To limit memory footprint, the shape model is trained during our first training stage supervised by GAN loss on the synthesized RGB images. At this stage, the spatial encoder is not used. This approach can be seen as a deformable variant of PiGAN. During stage 2, the shape model is fixed (thus is not updated) to save memory while the DNR is trained for photorealistic synthesis.
>
> **R: “Figure 7 … there is some high-frequency noise on the surface geometry … why is this happening? … Have the authors experimented with regularizing the shape which the TOCS represent?”**
>
> Answer: The high frequency noise in Paper Figure 7 occurs during the stage 1 shape training (see Paper Section 2.1). It is visible for the animal categories (AFHQ) but not for human faces (FFHQ) (see Paper Figure 1). It results from the jagged fur patterns found on the surface of the animals. For shape regularization, we followed NEUS, StyleSDF and used eikonal loss which we found to be essential - although costly for the memory budget to compute gradients in the forward pass. Under-regularization led to degenerate surfaces, whilst overregularizing caused oversmooth shapes which were harmful to FID scores.
> We agree that more research in this direction can be useful, but we argue that this is a common problem in current unsupervised 3D methods.

---

### Official Review · Reviewer_2TB7 · 2022-10-29

**Confidence:** 3
**Correctness:** 4
**Technical Novelty And Significance:** 3
**Empirical Novelty And Significance:** 3
**Recommendation:** 8

**Clarity, Quality, Novelty And Reproducibility:**

The paper is well-motivated and presented. A good attempt is made in this paper to bridge the gap between 2D images and 3D physical reality.

**Strength And Weaknesses:**

Pros:
1. It is a good idea to bride the 3D morphable models with GAN synthesis, enabling dense correspondence between generated objects.
2. The method is clearly explained and the text is well-written.

 I don't have major concerns about this paper. In general, I like the idea of 3D control of 2D generative models. A few minor comments are listed below.
1) Can the proposed model be applied to generic object image generation, such as cars, and buildings? Would the method work for objects with large intra-class deformations?
2) More qualitative and quantitative evidence of the 3D disentanglement is needed.

**Summary Of The Paper:**

The paper introduces a novel 3D-aware generative image model, StyleMorph, which can disentangle 3D shape, camera pose, appearance and background for high-quality image synthesis.  By bridging 3D morphable models with GAN synthesis and a canonical coordinate system, dense correspondences among generated objects can be provided. Experiments show disentangled control over pose, shape, object appearance, and background appearance for high-quality image synthesis.

**Summary Of The Review:**

Overall, I think the idea is interesting and the proposed "template object coordinates" is a good contribution in image synthesis.

---

> ### Author Response · Authors · 2022-11-19
> **Response to reviewer 2TB7**
>
> **R: “Can the proposed model be applied to generic object image generation, … cars and buildings? Objects with large intra-class deformations?”**
>
> Answer: We have added qualitative disentanglement results on car and architecture datasets in Supplemental Figures 10 and 11.
> Morphable models are not suitable for generic categories which include topological variation, like buildings or furniture, since such shapes cannot be easily expressed as smooth warps of a single template. Works such as [1], [2], [3] are emerging to address topological-variability in implicit shape models; we leave it to future work to incorporate such shape models into our photorealistic 3D aware synthesis task.
>
> [1] Generative Deformable Radiance Fields for Disentangled Image Synthesis of Topology-Varying Objects
>
> [2] Learning Implicit Functions for Topology-Varying Dense 3D Shape Correspondence
>
> [3]Topologically-Aware Deformation Fields for Single-View 3D Reconstruction
>
> **R: “More qualitative and quantitative evidence of the 3D disentanglement is needed.”**
>
> Answer: Please see Supplemental Figures 5, 6, 7, 9, 10, 11 (updated supplemental) for examples of expression / gender control via the deformation field, caricatures and results on additional datasets. As discussed in the section we do not want to claim we perform expression/gender control, but rather give some insights into how the shape space captures this information. Any other suggestions on qualitative results would be welcome.

---

### Official Review · Reviewer_XLzz · 2022-10-30

**Confidence:** 4
**Clarity, Quality, Novelty And Reproducibility:** 1. The writing is generally clear. Bu…
**Correctness:** 3
**Technical Novelty And Significance:** 3
**Empirical Novelty And Significance:** 3
**Recommendation:** 6

**Strength And Weaknesses:**

Strength:
1. It is an interesting paper that could generate 3D-aware and high-quality images with disentangled controls over 3D shape, camera pose, foreground appearance, and background appearance.
2. They utilize 3D morphable models for synthesizing high-quality images by introducing a canonical coordinate system, and therefore they could deform the 3D shape by morphing the 3D canonical template.
3. They show impressive synthetic images with disentangled controls.

Weakness:
The current manuscript seems to be a technical report, and the proposed method seems to be a comprehensive combination of existing methods.  I think it is an interesting work, but I am not sure if it is good enough for an ICLR publication.

**Summary Of The Paper:**

This work proposes a 3D-aware generative model to disentangle 3D shapes, camera poses, object appearance, and background appearance when synthesizing high-quality images, which could realize non-rigid shape variation in an object category exclusively from 2D image supervision.  To achieve a clear disentanglement, it proposes Template Object Coordinates (TOCS) to provide a powerful, deformation-equivariant descriptor of 3D shapes.  It provides impressive qualitative results with disentangled controls, and competitive quantitative results on the evaluated benchmark datasets.

**Summary Of The Review:**

In general, I think it is an interesting work, showing impressive results.  Although I am a little worried about its contributions and novelties, I tend to support its acceptance at the current stage.  I would reconsider my rating when the other reviews are available.

---

> ### Author Response · Authors · 2022-11-19
> **Response to reviewer XLzz**
>
> **R: “method seems to be a comprehensive combination of existing methods”**
>
> Answer: Please note that the state-of-the-art disentangled and deformable synthesis capabilities of our model (see Paper Table 1) are due to our usage of TOCS as a conditioning signal to the Deferred Neural Renderer (see Paper Ablation Table 2, Supplemental Ablation Table 2), a key novel contribution which is not found in any other 3D-aware GAN work.
>
> **R: “writing is generally clear…. research gaps … reads like a technical report”**
>
> Answer: We thank the reviewer for these suggestions, and have updated our manuscript accordingly. We have included Section 10 on Limitations (research gaps) in the Supplemental Material and reworded the contributions.
> We would appreciate more input on what makes the manuscript “read like a technical report” - we will try to integrate this in the final manuscript.
>
> **R: “contribution list could be reorganized …  current version not clear … kind of wordy … TOCS/ disentangled control could be summarized into one contribution”**
>
> Answer: Regarding condensing our list of contributions, we wanted to elaborate on two distinct aspects of TOCS: their definition as 3D functions that are mapping into template coordinates (established through a 3D deformation field), and their projection to 2D as a conditioning signal. In more standard graphics terms this would correspond to a deformable 3D surface and its UV mapping and the projection of this UV field to the image. We prefer to keep these separate as these are mathematically distinct.
>
> **R: “reference formats used in the manuscript are not consistent.”**
>
> Answer: We have updated the manuscript to the correct ICLR citation format.

---

### Official Review · Reviewer_J5vi · 2022-10-31

**Confidence:** 2
**Correctness:** 4
**Technical Novelty And Significance:** 3
**Empirical Novelty And Significance:** 3
**Recommendation:** 6

**Clarity, Quality, Novelty And Reproducibility:**

Quality: I think the paper achieves good quality and the proposed approach is interesting.
Clarity: I think there are some motivations that should be further introduced, e.g., why we need two stage training, why deformable field is really necessary?
Novelty: Good. I think the deformable field, the way to disentangle the four elements of image are interesting.
Reproducibility: Will you release code after publication? I think it is hard to implement the approach due to the complex pipeline and some missing details.

**Strength And Weaknesses:**

Strengths:
1. To disentangle the key elements such as camera pose, apperance et al. is a important research topic, and the proposed approach is interesting and achieves good performance on this topic. Some illustrated results are impressive.
2. Most of the quantitative results outperform existing works, demonstrating the superiority of the proposed approach. The experiments are conducted on four datasets, which are sufficient.

Weaknesses and questions:
1. Some works like [1] provide more than the four choices in this paper for manipulation, like illumination, expression et al, while this work only covers the four(camera pose, back gournd, appearence, shape). In other word, I think the shape/appearance manipulation cannot support fine-grained manipulation, and this paper lacks detailed manipulation capability.
[1] A. Tewari et al, StyleRig: Rigging StyleGAN for 3D Control over Portrait Images, CVPR 2020
2. The quality of the reconstructed 3D shapes seems coarse compared with EG3D [2], i.e., Figure 2 in supplementary file.
[2] ER Chan et al. EG3D: Efficient Geometry-aware 3D Generative Adversarial Networks, CVPR 2022
3. (1) Is TOCS map provide all necessary information for stage 2 training?
(2) How about train the whole framework end-to-end?
(3) How does the GAN loss in stage 1 helps stage 2 trainin? It seems only TOCS goes into stage 2 but the generated image in stage 1 not goes to stage 2.

**Summary Of The Paper:**

This paper introduce a 3D-aware generative model that can control 3D shape, camera pose, object appearance and background independently. The author connect 3D morphable modelling with deferred neural rendering by performing an implicit surface rendering of TOCS, and construct a purely geometric, deformation-equivariant 2D signal that reflects the compounded geometric effects of non-rigid shape, pose, and perspective projection. The experiments of 4 datasets shows the good disentanglement capability of the proposed approach.


**Summary Of The Review:**

Overall, I think it is a good paper. Though I am not experted in 3D-aware GANs, some results in the paper impressed me and I tend to give positve rating to this paper.

---

> ### Author Response · Authors · 2022-11-19
> **Response to reviewer J5vi**
>
> **R: “Some works like [1] provide more than the four choices for manipulation … this work only covers pose, background, appearance, shape … this paper lacks detailed manipulation capability."**
>
> Answer: Comparison with [1]: The main difference with StyleRig [1] is that [1] requires a pretrained parametric 3D morphable face model to provide a conditioning signal (Figure 1 of [1]). This provides mesh face shapes with per-vertex colour information, and of course requires hard-to-obtain 3D human face scans to train.
> In contrast, our work learns an implicit morphable model from raw images without any 3D supervision or prior, and is therefore applicable to general image categories beyond just human faces (as shown by our results e.g. the AFHQ datasets).  Our model’s manipulation capabilities and synthesis quality are state-of-the-art when compared with other unsupervised methods as shown in Paper Table 1. We agree that controlling illumination is an interesting extension, while some form of image-level supervision could also allow us to control expression - we will pursue such directions in future work.
>
> **R: “quality of the reconstructed 3D shapes seems coarse compared with EG3D [2]”**
>
> Answer: The reason for not using EG3D is purely because of hardware limitations: we are memory-limited in the resolution at which we can render the geometry during training. Using 2 implicit networks for shape (template occupancy and deformation-offsets) does increase the GPU memory footprint, but allows us to register all generated shapes to the template. We only have access to 4 Tesla V100 GPUs for training - therefore we render the TOCS maps at resolution 64 (final paragraph of Paper Section 2.1). In contrast, in EG3D the model is trained using 8 V100 GPUs, and does not use a separate deformation network, allowing them to render their implicit field at resolution 128.
> We anticipate that hardware advances will remove this problem in the coming years - but do not believe this should be counted against our work.
>
> **R: “(1) Does TOCS provide all necessary information for stage 2? (2) train end-to-end? (3) How does GAN loss in stage 1 help stage 2 training? TOCS goes into stage 2 but the generated image in stage 1 does not go to stage 2.”**
>
> Answer: Yes, the TOCS map is the only input to the DNR during stage 2. Due to GPU memory constraints we are prevented from training the system end-to-end. During the full-resolution stage-2 training, we don’t have sufficient GPU memory budget to pass gradients through our Volume Renderer’s implicit networks. We therefore train the Volume Renderer in a separate low-resolution training stage (see Paper Section 2.1), which is standard amongst 3D-aware GAN works (e.g. StyleNeRF, EG3D and StyleSDF). Regarding “How does the GAN loss in stage 1 helps stage 2 training?” The low-resolution RGB and GAN loss are only used for learning the shape model during stage 1, which is then frozen for stage 2.
>
> **R: “some motivations should be further introduced, e.g., why two stage training, why deformable field?”**
>
> Answer: The deformable field allows us to learn a shape model which registers all generated shapes to a template, providing surface-level correspondences between samples. This facilitates 3D geometry-based control, which is crucial to graphics-related applications such as Augmented Reality. The deformation field also enables us to use TOCS maps (Object Coordinates in Template space) as a strong deformation-equivariant shape conditioning signal for optimal performance as shown in ablations (Paper Table 2). We address this in the updated manuscript.
>
> **R: “Reproducibility: Will you release code after publication?”**
>
> Answer: Yes, we promise to release the code upon publication.

---

### Author Response · Authors · 2022-11-19
**General response**

We thank the reviewers for their insightful comments and helpful feedback. We are glad that all reviewers appreciated the quality of our results and that R1-R4 found our approach novel.

As requested, we have updated the supplemental with multiple new results to provide further evidence for the merit as well as generality of our method.
These include 6 additional results pages (page 5-10), comprising TOCS-related ablations (Table 2), expression control (Figs 5,6), caricatures (Fig 9) and experiments on new, challenging categories (Figs 10 & 11), among others. We hope these additional results will provide sufficient evidence to address concerns raised by the Reviewers.

We have further updated our manuscript, following the Reviewer suggestions, to clarify the methodology and cite additional related works.

We now provide a point-by-point reply to individual Reviewer questions and concerns in the comments below.

---

### Decision · Program_Chairs · 2023-01-20

**Decision:**

Accept: poster

**Justification For Why Not Higher Score:**

There are significant presentation issues in particular regarding the novelty claims

**Justification For Why Not Lower Score:**

The results are strong and the TOCS formulation may have some value

**Metareview: Summary, Strengths And Weaknesses:**

This submission introduces a 3D generative model with a disentangled representation of shape, pose, and texture for image synthesis. Reviewers in general like the formulation and the good results; however, there are concerns about the novelty and the inappropriate overclaims in the paper. There is a lot of discussion about the paper post-rebuttal, with four reviewers being positive and one being negative.

The AC has been following the discussion and also invited two external evaluators who work exactly on this topic to provide additional feedback. The consensus is that this paper has value in terms of its TOCS formulation and its good results. However, current overclaims are inappropriate, related works should be better explained and novelty claims should be revised, and additional ablation studies especially on the benefits of TOCS would certainly be helpful.

The AC decides to recommend acceptance due to the potential value of the paper, but urges the authors to revise the paper accordingly.  In particular, reviewer cRkf provided the following candid feedback during the discussion, which the AC, reviewers, and additional evaluators all agree on.

"The main concerns are the novelty.

* Morphable Renderer Similar to Disentangle3D[1]. Other reviewers mainly praise the disentanglement of this paper. However, the Morphable Renderer, which accounts for the disentanglement is technically similar to the Disentangle3D[1], a CVPR'22 paper. I point this out as the first point of weakness and the authors did not tell the difference between the Morphable Renderer and Disentangle3D[1]. Instead, they strengthen that their image quality is better, which is misleading because the enhancement of image quality comes from the Deferred Neural Renderer, rather than the Morphable Renderer. In my opinion, in order to demonstrate their novelty on Morphable Renderer, they ought to:

1. directly tell the difference between the Morphable Renderer and Disentangle3D[1],
2. report the FID of the first stage output of the Morphable Renderer to show the superiority of the Morphable Renderer in terms of image quality.
* Deferred Neural Renderer Similar to StyleNeRF[2], EG3D[3], etc. The Deferred Neural Renderer is quite common for the NeRF-based generative model to enhance image quality. Although there might be some differences in implementation, I think the main idea is similar. And the use of TOCS as the super-resolution input is too small a point to be the main contribution of an ICLR paper.

In rebuttal, the authors go around that Disentangle3D[1] has poor image quality while other high-res methods are not disentangled, implying this work is actually a combination of Disentangle3D[1] and a super-resolution module. "


**Note From Pc:**

if the above contains the word "oral" or "spotlight" please see: "oral" presentation means -> notable-top-5% and "spotlight" means -> notable-top-25%. As stated in our emails, we are disassociating presentation type from AC recommendations